# CryoEM reveals how the complement membrane attack complex ruptures lipid bilayers

Anaïs Menny[1], Marina Serna[1,2], Courtney M. Boyd[1], Scott Gardner[1], Agnel Praveen Joseph[3,6], B. Paul Morgan[4], Maya Topf [3], Nicholas J. Brooks [5] & Doryen Bubeck [1]

The membrane attack complex (MAC) is one of the immune system's first responders. Complement proteins assemble on target membranes to form pores that lyse pathogens and impact tissue homeostasis of self-cells. How MAC disrupts the membrane barrier remains unclear. Here we use electron cryo-microscopy and flicker spectroscopy to show that MAC interacts with lipid bilayers in two distinct ways. Whereas C6 and C7 associate with the outer leaflet and reduce the energy for membrane bending, C8 and C9 traverse the bilayer increasing membrane rigidity. CryoEM reconstructions reveal plasticity of the MAC pore and demonstrate how C5b6 acts as a platform, directing assembly of a giant β-barrel whose structure is supported by a glycan scaffold. Our work provides a structural basis for understanding how β-pore forming proteins breach the membrane and reveals a mechanism for how MAC kills pathogens and regulates cell functions.

[1] Department of Life Sciences, Sir Ernst Chain Building, Imperial College London, London SW7 2AZ, UK. [2] Spanish National Cancer Research Centre, CNIO, Melchor Fernández Almagro, 3.28029 Madrid, Spain. [3] Institute of Structural and Molecular Biology, Department of Biological Sciences, Birkbeck, University of London, Malet Street, London WC1E 7HX, UK. [4] Division of Infection and Immunity, School of Medicine, Cardiff University, Heath Park, Cardiff CF14 4XN, UK. [5] Department of Chemistry, Imperial College London, London SW7 2AZ, UK. [6] Present address: Scientific Computing Department, Science and Technology Facilities Council, Research Complex at Harwell, Didcot OX11 0FA, UK. These authors contributed equally: Marina Serna, Courtney M. Boyd. Correspondence and requests for materials should be addressed to D.B. (email: d.bubeck@imperial.ac.uk)

The multiprotein membrane attack complex is a bactericidal weapon of the innate immune system that also modulates inflammation and proliferation when formed on self-cells[1]. The MAC pore targets a wide range of pathogens, forming on and killing Gram-negative bacteria[2], enveloped viruses[3], and parasites[4]. This innate immune effector is essential for fighting bacteria of the genus *Neisseria*;[5] with genetic deficiencies in component proteins leading to recurrent infections[6]. Host cells are protected from bystander damage by the GPI-anchored receptor CD59, the only membrane-bound inhibitor of MAC[7]. Deficiency of CD59 causes the lethal blood disorder Paroxysmal Nocturnal Hemoglobinuria, a disease characterized by thrombosis and chronic hemolysis[8]. Complement activation and MAC formation also contributes to killing of cancer cells during immunotherapy treatments[9]. Therefore, developing a molecular model of how MAC forms on target cells is essential for understanding the immune response to microbes and for the development of therapeutics that regulate complement activity.

MAC assembles from soluble complement proteins in an obligate sequential pathway[10]. In the presence of membranes, C5b6 (a complex comprised of C5b and C6[11]) binds C7 to form the lipophilic MAC precursor C5b7[12]. C8 irreversibly binds the nascent complex, resulting in the membrane-inserted C5b8[13]. C9 molecules associate with C5b8 in the membrane to form C5b9 and polymerize to complete the MAC pore[14]. Previous electron cryo-microscopy (cryoEM) reconstructions of the complex revealed that the final MAC is comprised of 6 polypeptide chains (C5b, C6, C7, C8α, C8β, and C8γ) together with 18 C9 monomers that are arranged in a split-washer configuration[15,16]. Based on structural similarity with bacterial homologs, the giant (110 Å diameter) β-barrel pore is formed when helical bundles in the Membrane Attack Complex-Perforin (MACPF) domains transform into transmembrane β-hairpins (TMH)[17–19], although a molecular mechanism for how this occurs is not currently understood.

In this study, we used cryoEM to determine the structures of two MAC conformations at near atomic resolution and derive a nearly complete atomic model for the pore. In combination with flicker spectroscopy, we show how MAC assembly impacts biophysical properties of the bilayer and resolve the mechanisms of membrane interaction and MAC activity.

## Results

**CryoEM structure of the MAC.** The human MAC pore was formed on liposomes from individual complement proteins. The lipid composition of vesicles was selected based on the stoichiometric homogeneity of deposited pores. MAC was solubilized with detergent and purified for structural studies, as described previously[15]. By integrating newly collected data across multiple electron microscopes (Supplementary Table 1), we were able to improve the overall resolution of the MAC from 8.5 Å[15] to 4.9 Å; however, density corresponding to the interface between C6 and the terminal C9 was still poorly resolved. We used 3D classification procedures to computationally isolate two stoichiometrically identical conformations, open and closed, which varied in the extent of β-barrel closure (Fig. 1 and Supplementary Fig. 1). 2D classification of negatively stained complexes inserted into a lipid monolayer confirmed the presence of these states in a membrane environment (Supplementary Fig. 2a), in agreement with cryotomography structures of MACs in liposomes[16]. The maps were further subdivided into three components: an asymmetric region (C5b, C6, C7, and C8), a hinge region (C7, C8, and two C9 molecules), and a C9 oligomer. Using a masked refinement strategy coupled with signal subtraction[20], we improved the resolution of the asymmetric regions for each conformation to

4.7 Å and 5.9 Å (Fig. 1, Supplementary Figs. 1, and 2b). The hinge region of the open conformation was resolved to 4.9 Å (Supplementary Figs. 1 and 2b). Masked refinement from signal-subtracted images followed by sub-volume averaging was used to resolve the averaged C9 monomer from the open conformation at 4.4 Å (Fig. 1a, and Supplementary Figs. 1, 2b, 3c). A similar analysis of the C9 oligomer from the closed conformation resulted in a lower resolution map. Therefore, we focused our interpretation of C9 on density derived from the open conformation oligomer. The new maps enabled us to build an atomic model that includes the irregular and asymmetric β-barrel pore (Supplementary Table 1 and Supplementary Fig. 3). Although density is lacking for many side-chains within C5b and the lower half of the central β-barrel, we have imposed experimental restraints that justify their register in the atomic model. Crystal structures for soluble components (C5b6[11] and C8 [21], a heterotrimeric complex consisting of α, β, and γ polypeptide chains) together with homology models for C7 and C9 were fitted into the density. Domains of these structures were first refined as rigid bodies, with disulfide bond restraints. Models were further refined restraining secondary structure and side-chain geometry to higher resolution crystallographic structures. β-strands that comprise the central barrel were initiated where side-chain density was visible and extended imposing idealized backbone geometry constraints. The trajectory for each strand is linear and the register was confirmed by correlating glycan density with the position of the modified residue in the sequence (Supplementary Fig. 4).

**MAC is a flexible immune pore.** We used 3D classification to resolve MAC conformational flexibility (Fig. 1 and Supplementary Fig. 1). The open conformation is characterized by a 30 Å wide chasm that runs the length of the complex (Fig. 1b). Lipid molecules likely fill the opening on the wall of the pore, reminiscent of arc pores observed for both mammalian and bacterial β-pore forming proteins[22,23]. The asymmetric region juts into the lumen of the barrel like a "paddle", exaggerating the MAC's split washer shape (Fig. 1b and Supplementary Movie 1). C8γ is wedged in the crease between the rotated asymmetric component and adjacent C9 oligomer, and may limit the rotation of the paddle. While curvature of C9 arcs vary at either end, the central section is near-circular with monomers equally spaced ~16° apart (Fig. 1b, c), reminiscent of the arrangement observed for a C9 homo-oligomer[24]. Although the chasm is sealed in the closed conformation (Fig. 1c), interfaces mediating the MACPF-rim and transmembrane regions are not flush. The asymmetric region swings back and meets C9 in a noncanonical MAPCF-thrombospondin (C6-TSP3) interaction with limited buried surface area (Fig. 1d). Despite a contiguous extracellular β-barrel, there remains a gap within the transmembrane pore where the shorter hairpins of C6, C7, and C8 abut those of C9 (Fig. 1c). In the open conformation, the first and terminal C9 are latterly shifted by ~20 Å, while the ring of the closed conformation remains planar (Fig. 1). Therefore, conformational flexibility of the assembly may impact local curvature of the membrane.

The MAC is a highly-glycosylated assembly with all complement components post-translationally modified[11,25–27]. We observe density for many of the reported glycans on C7[25], C8[27], and C9[26], which line the β-barrel's concave face (Figs. 1b, c, 2a and Supplementary Fig. 4). Glycan removal led to irregular pores with significantly distorted curvature (Supplementary Fig. 4). These data suggest glycans could play a role in maintaining the structural integrity of a flexible giant β-barrel. Although deglycosylation did not impact the ability to rupture simple model membranes (Supplementary Fig. 4), glycans may confer greater robustness on pore assembly.

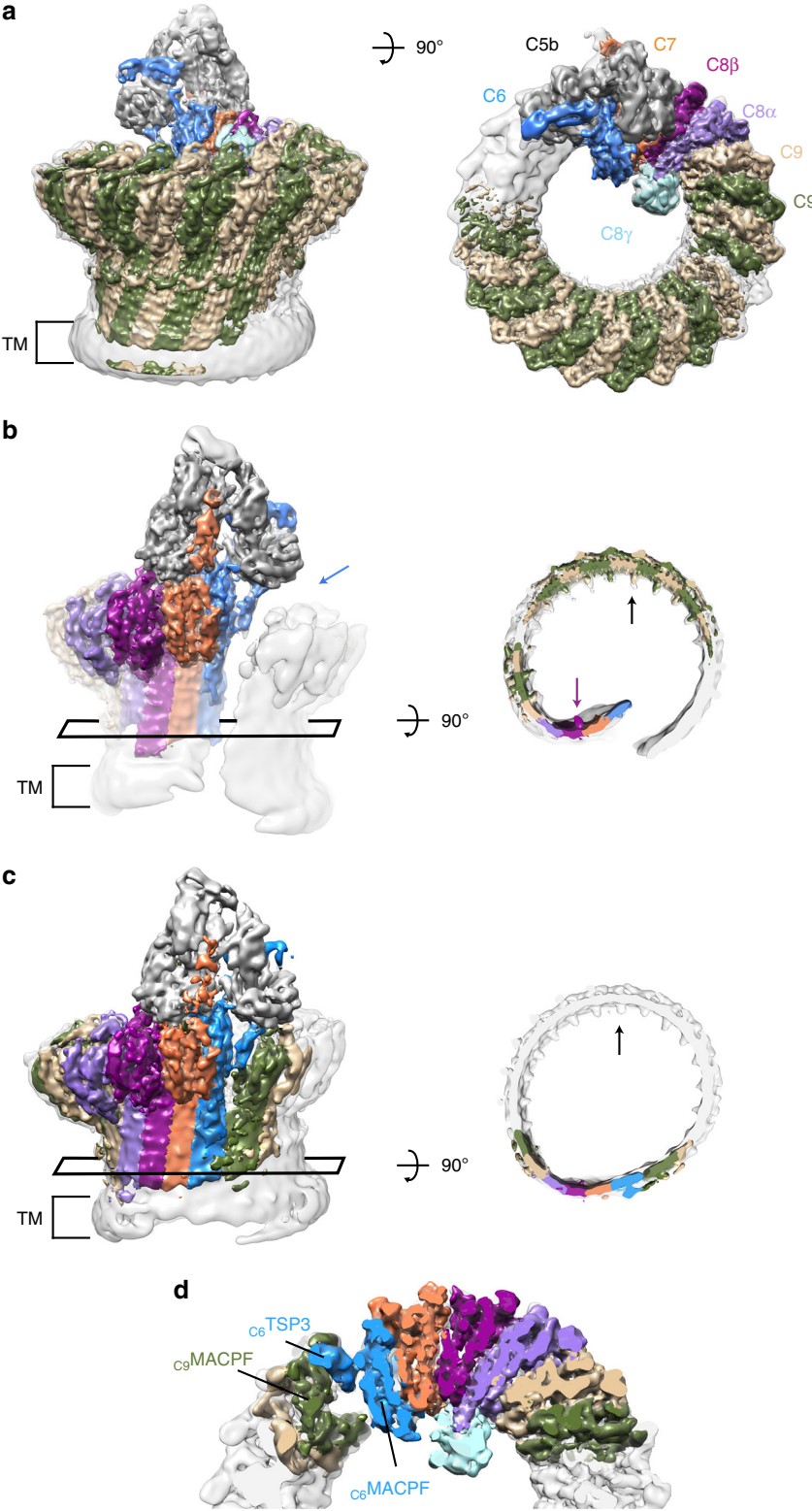

**Fig. 1** CryoEM structures of MAC reveal conformational flexibility of the pore. **a** CryoEM density map for the open conformation (gray transparent surface) overlaid with subvolume reconstructions of the asymmetric, hinge and C9 oligomer regions (colored according to protein components). Transmembrane region is indicated (TM). **b** CryoEM reconstructions in **a** rotated 180°. Blue arrow indicates the gap between C6 and terminal C9 of the split washer. A rectangle highlights a cross-section of the MAC β-barrel shown in the right panel. Black arrow highlights density for C9 glycans that protrude into the lumen of the barrel. Purple arrow indicates density for N-linked glycosylation of C8β (N189) within the hinge. **c** CryoEM map of the closed conformation (gray transparent surface) overlaid with the subvolume reconstruction of the asymmetric region (colored density) in the same orientation as **b**. Rectangle indicates the cross-section shown in the right panel. **d** Closed conformation of the MAC slabbed through the core MACPF domains. C6 TSP3 and MACPF domains of C6 and C9 are indicated

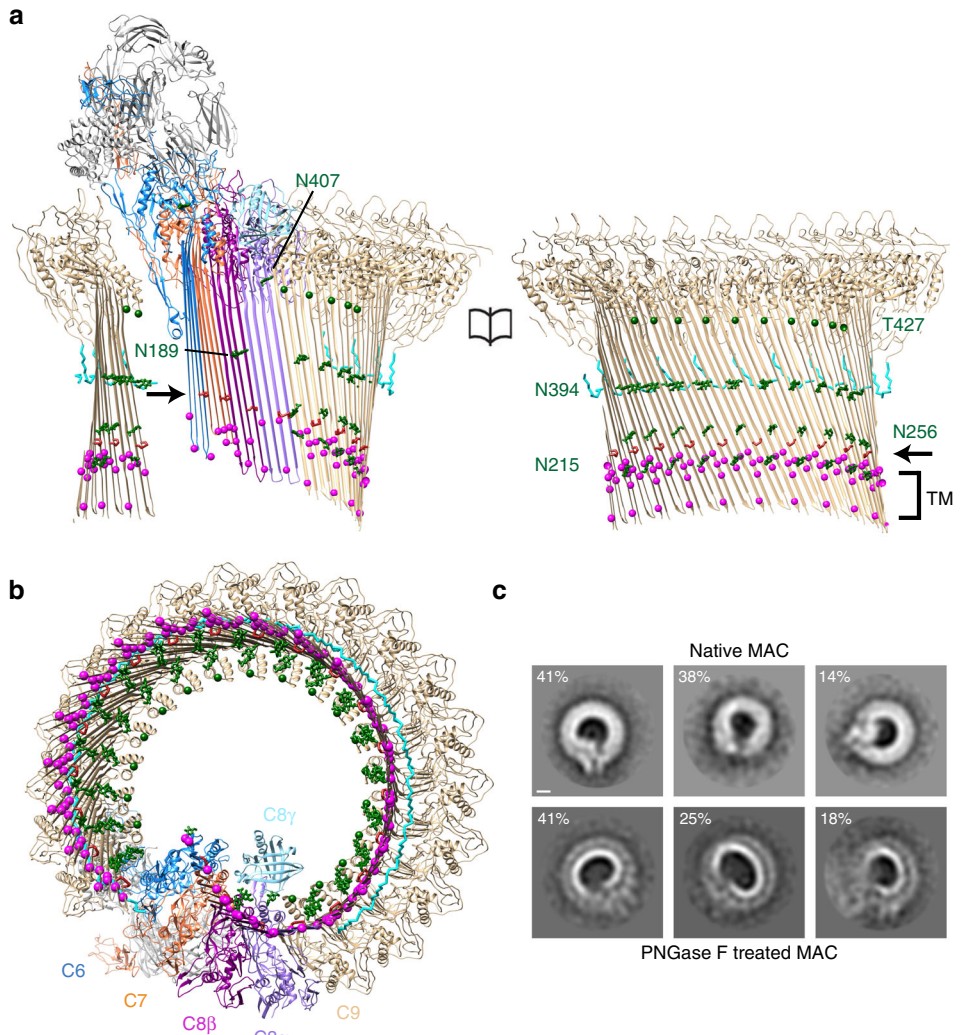

**Fig. 2** Glycans provide structural support for a giant β-barrel. Molecular model for the open conformation as viewed from within the β-barrel (**a**) and from the membrane (**b**). N-linked glycans (green) are shown as sticks (C9:N215, N256, N394; C8β:N189; and C8α:N407). Cα carbons of O-linked C9:T427 are represented as green spheres. Backbone trace of C9 C-terminus (res 522:535) is shown as cyan sticks. Disulfide bonds within MACPF domains are red and indicated by an arrow. Cα carbons of aromatic residues near the membrane (TM) are magenta spheres. **c** Reference-free 2D class averages of negatively stained MAC assembled from de-glycosylated C8 and C9 (bottom panel) show distorted pores on lipid monolayers as compared to native MAC (top panel). Classes are derived from similar particle numbers to enable a more direct comparison of curvature distortion (native, 706; de-glycosylated, 711). Native MAC particles are a randomly chosen subset of the data shown in Supplementary Fig. 2a which include variations across open and closed conformations. Percentage of particles belonging to each class is indicated. Scale bar, 50 Å

**C5b6 is a spatial platform for MAC assembly**. Limited resolution of previous MAC reconstructions[15,16] prevented any structural analysis of rearrangements within C5b6 during pore formation. We therefore investigated how C7-binding to C5b6 triggers the lipophilic transition using our high-resolution maps (Fig. 3). Our data reveal that the C5b6 complex serves as a spatial platform directing MAC assembly. C5b MG domains 1, 4, and 5, together with the "link" domain, bridge a cluster of arches comprised of the lipoprotein receptor class A (LDL) domains of C6, C7, and C8β (Fig. 3a and Supplementary Fig. 5). While the core of C5b remains largely unchanged during the transition, C6 undergoes marked domain rearrangements upon integration into the MAC.

C6 is comprised of 10 individual domains that can be classified into three functional parts: (1) those that mediate the interaction with C5b, (2) regulatory auxiliary modules, and (3) the pore-forming MACPF domain. Large structural re-arrangements of C6 auxiliary domains accompany conformational changes within the pore-forming MACPF (Fig. 3d and Supplementary Fig. 5a).

Superposition of the soluble and MAC-incorporated forms of C5b6 show that although the relative orientation of the C5b thioester-like domain (TED) and C6 C-terminal complement control protein (CCP) domains remains unchanged, the C6 LDL is displaced by C7-binding. The two N-terminal TSP domains (TSP1 and TSP2) undergo a concerted rotation with respect to the core MACPF, resulting in a final position near-perpendicular to the plane of the membrane, stabilizing the newly formed β-sheet. Rotations of regulatory auxiliary modules coincide with an unbending and untwisting of the C6 MACPF β-sheet. Movement of the C6 epidermal growth factor (EGF) domain and MACPF helix-turn-helix (CH3) motif release the pore-forming TMH regions, in agreement with lower resolution structures of pore-forming toxin homologs[28].

**Auxiliary domains mediate the lipophilic transition**. We next explored whether conformational changes incurred by C6 were

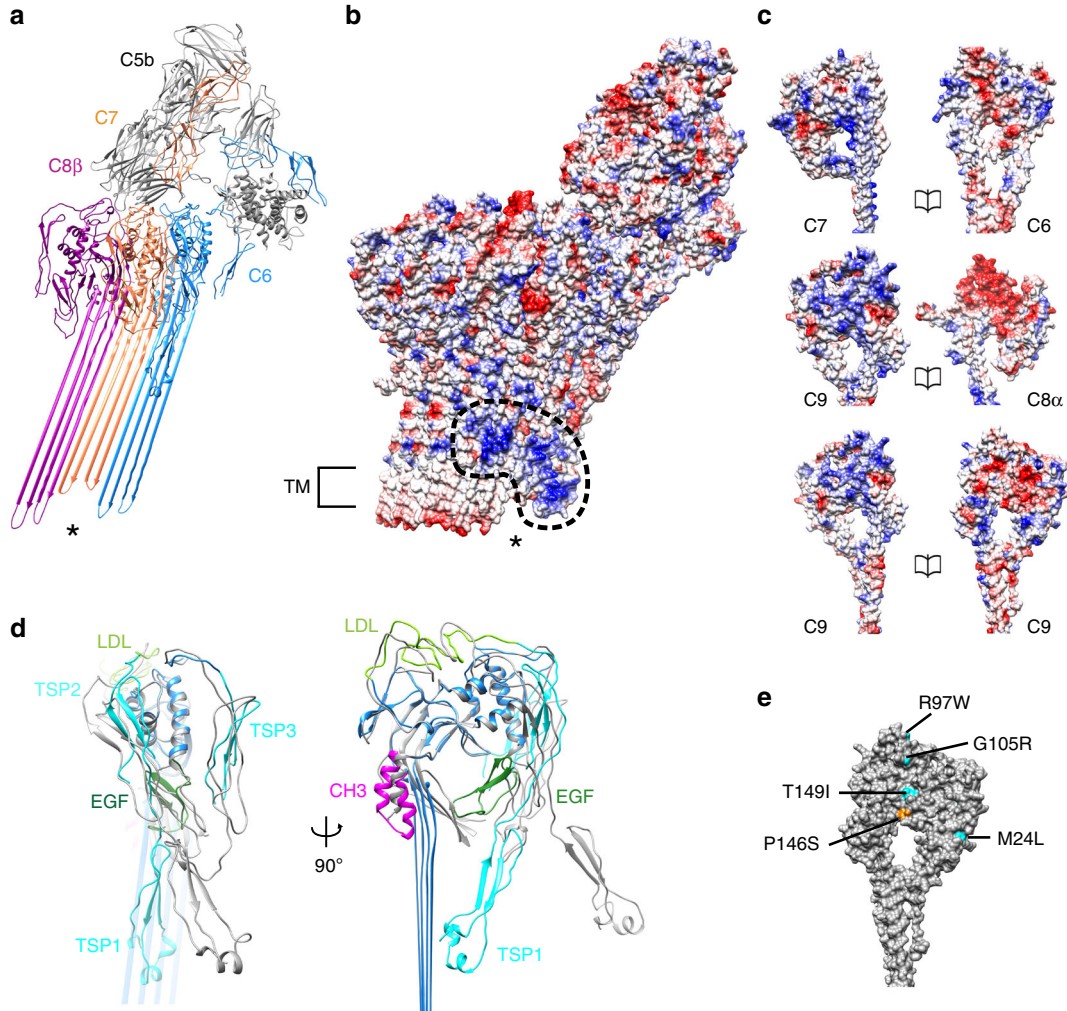

**Fig. 3** Structural transitions of complement proteins upon MAC assembly. **a** Molecular model of complement proteins that interact with C5b (gray).
\* indicates uneven lengths of TMH regions of C6 (blue), C7 (orange), and C8β (magenta). **b** Coulombic surface potential (from red -10 kcal/mol*e to
blue +10 kcal/mol*e) of atomic models for C5b, C6, C7, C8 and two adjacent C9 monomers viewed from the convex face of the barrel (same view as in **a**).
Dotted line highlights a patch of positively charged residues within the membrane-interacting regions of C6, C7, and C8. Transmembrane regions of C8
and C9 are indicated (TM). \* indicates the same region as in **a**. **c** Coulombic surface potential representation for MACPF interaction interfaces of
complement proteins. **d** Superposition of the MAC form of C6 (colored) and C6 from the soluble C5b6 crystal structure lacking TMH residues (PDB:
4A5W; gray). Rotations in C6 TSP (cyan), LDL (light green), and EGF (dark green) domains accompany movement of CH3 latch (magenta) and unfurling
of transmembrane β-hairpins. **e** Genetic variants identified in AMD patients mapped on the C9 structure. Mutations reported to decrease polymerization
are cyan, those reported to cause spontaneous self-polymerization are orange[30]

conserved across complement proteins. N- and C-terminal TSP domains of C7, C8 (TSP1 and TSP2), and C9 (TSP1), overlay with those of C6 (TSP2 and TSP3) in its membrane-inserted "active" conformation (Supplementary Fig. 6a). Furthermore, the position of CH3 relative to the MACPF β-sheet is the same for all MAC components (Supplementary Fig. 6b). Although the core MACPF domains of all MAC components overlay well, the glycine-rich hinge enables a wide range of angles to accommodate the varied curvature of the barrel. Similar to the pore transition of C6, C8 regulatory auxiliary modules (EGF and TSP2) release TMH regions of both α and β chains (Supplementary Fig. 6c). However, unlike the dramatic C6 MACPF unbending, the angle of the C8β MACPF sheet remains constant (Supplementary Fig. 6d). Component strands untwist to align with the C7 β-sheet and pack against the C7 CH3 latch. Surprisingly, C8β CH3 does not undergo a lateral shift during pore formation, suggesting that the C8β MACPF is already primed for membrane insertion.

To investigate the molecular basis underpinning the directionality of MAC's sequential assembly, we compared electrostatic surface potentials of complement proteins (Fig. 3c). The interface between C6-C7, C8α-C9, and C9-C9 is formed mainly by polar and charge interactions (Supplementary Table 2). Incorporation of C8α exposes a negatively charged patch that may influence recruitment of the positively charged face of C9. C9 oligomerization could be propagated by electrostatic complementarity between leading and lagging faces, consistent with CDC pore formation[29]. Mutations that alter polymerization of C9 are implicated in some cases of age-related macular degeneration (AMD)[30]. We are now able to assign disease-related variants on the human C9 structure, and find that three of these AMD-associated mutations (P146S, G105R, and T149I) would likely impact the negatively charged patch that drives oligomerization. G105R and T149I variants decrease polymerization of C9[30] and would reduce the footprint of the negatively charged interface. By

contrast, variant P146S increases self-polymerization of C9[30] and may act by increasing polarity of the surface.

**Interactions with the lipid bilayer**. Complement proteins within the MAC interact with the lipid bilayer in two distinct ways. Transmembrane hairpins of MAC components vary in length and amino acid composition. C6 and C7 hairpins contain a single band of aromatic residues (Fig. 2a). A patch of positively charged residues resides proximal to the tips of the short C6 and C7 β-hairpins (Fig. 3a, b), suggesting interaction with phospholipid headgroups of the outer leaflet. Similar to other β-barrel pore assemblies[31], two rings of aromatic residues separated by the width of the bilayer anchor C8 and C9 within the membrane (Fig. 2a). The longer hairpins of C8 and C9 expose hydrophobic residues on the β-barrel's outer surface (Fig. 3b), consistent with an interface with lipid tails.

To investigate how these two types of interactions impact mechanical properties of the target cell membrane, we performed fluctuation analysis (flicker spectroscopy) of giant unilamellar vesicles (GUVs). In flicker spectroscopy, the magnitude of thermally induced fluctuations in a GUV membrane are quantified by tracking the membrane contour in a series of video-microscopy images. This data is fitted to a two-parameter model to extract the membrane bending rigidity and tension[32]. We used phase contrast light microscopy to track individual GUVs at each step of MAC assembly and confirmed the presence of functional pores by diffusion of sucrose across the bilayer (Fig. 4a). We recorded membrane fluctuations after the addition of C5b6, C7, C8, and C9, or when buffer replaced C5b6 in the sequence. When C5b6 was absent, the amplitude of the GUV membrane fluctuations remained constant throughout the experiment (Fig. 4a, b and Supplementary Movie 2). The small increase in fluctuations observed at later time points is likely to be due to slight increases in temperature caused by lamp heating. By contrast, membrane fluctuations of GUVs that supported MAC assembly were reduced upon C8-binding and were diminished to negligible levels with the addition of C9 (Fig. 4a, b and Supplementary Movie 2). These data suggest that large MAC lesions in cells may lead to rigidification of the membrane by the insertion of β-hairpins across the bilayer and by equilibration of the extracellular milieu across the pore.

Where fluctuation amplitudes could be parameterized, we analyzed changes in bending modulus and tension during MAC assembly (Fig. 4c and Supplementary Fig. 7). Bending modulus is an intrinsic property of the membrane and describes the amount of energy required to change the mean curvature of a lipid bilayer. It can be affected by thickness of the bilayer[33], membrane asymmetry[34], and lipid composition[35]. C5b6 is a soluble complex that ionically associates with membranes[36]. Upon C7-binding, the complex becomes lipophilic and is irreversibly tethered[12]. Although this study and previous work are consistent with an interaction of C7 with lipids[12,37,38], the ability of C5b6 to alter the biophysical properties of the membrane was unanticipated. C5b6 reduced the bending modulus of GUVs without affecting tension (Fig. 4c and Supplementary Fig. 7). The reduction in bending modulus was maintained upon C7 binding and anchoring of the C5b7 complex to the GUV. As C8 incorporates into the MAC to form C5b8, bending modulus increases (Fig. 4c). By comparison, tension remains largely unchanged (Supplementary Fig. 7a). Taken together, MAC specifically impacts the energy required to bend the membrane in a way that changes during the sequential assembly pathway. While our experimental system assumes a uniform distribution of pores on GUVs, MAC formation on target cells is heavily influenced by activation of upstream complement pathways and opsonization of bacterial cells. These

deposition hotspots may therefore influence changes in bending rigidity coefficients in a concentration-dependent manner.

**Discussion**

We have presented here cryoEM structures of two MAC conformations and derived an atomic model for the transmembrane complex. Furthermore, we have shown corroborating flicker spectroscopy data that explain how MAC assembly precursors alter physical properties of the membrane to prime the bilayer for attack. Our results provide a structural framework on which to interpret earlier biological data and inform future mechanistic models. Our structure has revealed a number of features that increase our understanding of the complement system, and informs a general mechanism for how β-barrel pore-forming proteins rupture lipid bilayers.

Previous high resolution structures of β-pore forming proteins were based on the assumption that the oligomeric assemblies are both rigid and symmetric[29,39], however we discovered that these proteins can be flexible within the membrane. Here we demonstrate that complement proteins within the C5b8 paddle rotate with respect to the nascent C9 oligomer, which in itself varies in curvature between the two observed MAC conformations (Supplementary Movie 1). A glycan scaffold that lines the inner wall of the giant β-barrel constrains the range of movement (Fig. 2). Together with C8γ, wedged between C5b8 and C9, this scaffold provides a structural support that maintains the curvature of an otherwise unsupported tall and flexible giant β-barrel pore. We propose that rotation of pore β-hairpins within the bilayer could provide an additional level of membrane destabilization that contributes to lytic activity of β-pore forming proteins. Bacterial membranes are complex targets, densely packed with both polysaccharides and porins[40]. Flexibility of a growing MAC pore would allow short stretches of β-hairpins to move within the plane of the bilayer to accommodate the compositional complexity of its membrane environment.

One of the most striking observations from our structure is that the MAC pore is comprised of β-hairpins that differ in both length and charge properties. These differences provide the structural basis for how complement proteins interact with target cells. Partially inserted β-hairpins of C6 and C7 interact with lipid headgroups of the outer leaflet and decrease the bending modulus of the membrane (Fig. 4). Subsequently, the increase in membrane stiffness we observe upon C8 addition could be attributed in part to the mechanical strain of distorting the bilayer. Membrane interacting β-hairpins of C6, C7, and neighboring C8β differ in length (Fig. 3a) and likely bend the bilayer to form the non-lamellar edge of an arc-pore. Levels of MAC on the plasma membranes of self-cells are restricted by endocytic pathways mediated by changes in the mechanical properties of the lipid bilayer[41,42]. This mechanism plays an important role in cell activation pathways[41], but also in the immune response to cancer[9]. In our model, we propose that differences in charge and length of the pore β-hairpins impact physical properties of the lipid environment that could stimulate activity of mechanosensors, such as caveolin-1 to facilitate MAC removal and recovery from MAC attack. Similarly, membrane repair mechanisms triggered in response to bacterial pore-forming toxins may rely on changes in mechanochemical properties of the plasma membrane caused by oligomerized prepores[43]. Although pore forming hairpins of these bacterial homologs extend throughout the bilayer, we speculate that an auxiliary domain's cholesterol recognition loops that interact with the outer leaflet may serve a similar role as the partially-inserted MAC precursor.

Whereas this study focuses on how MAC interacts with the membrane, C8 and C9 also bind CD59 to block complement-

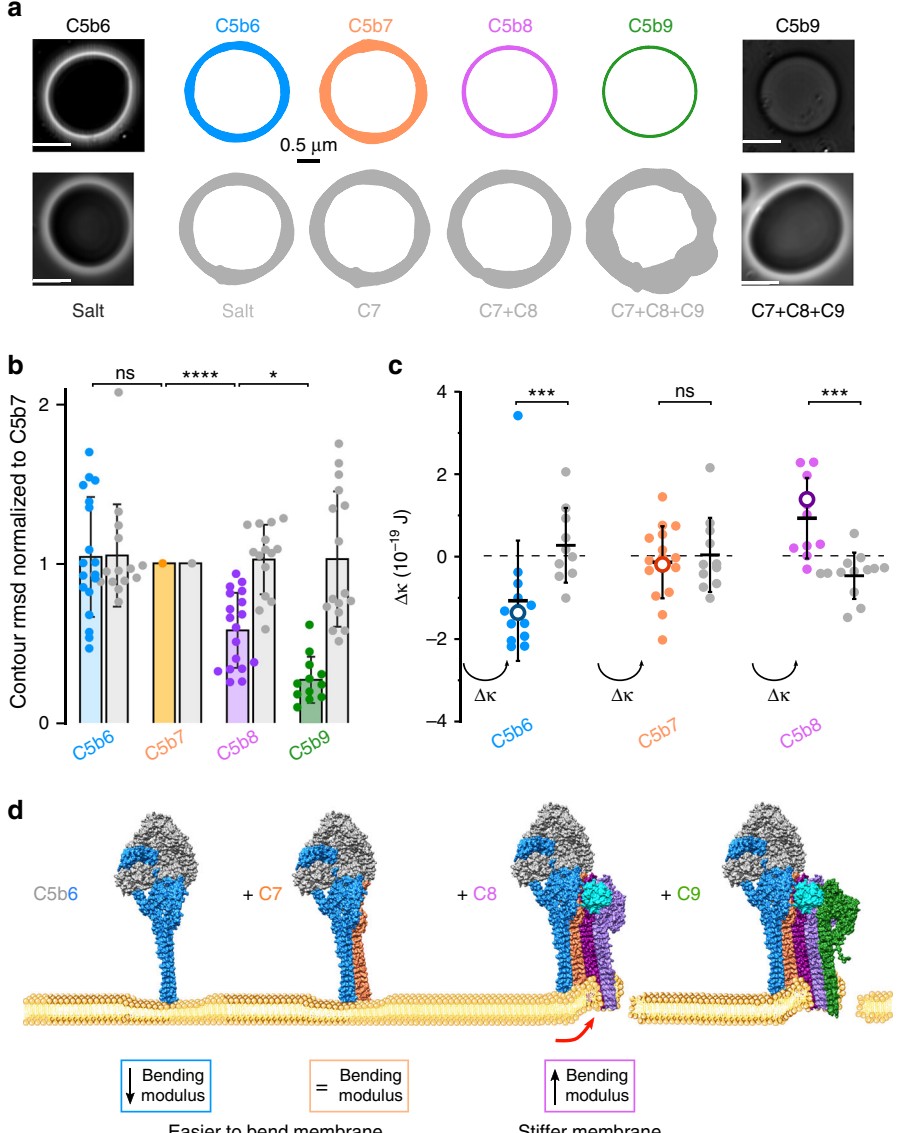

**Fig. 4** Flicker spectroscopy informs a biophysical timeline of pore formation. **a** Phase contrast light microscopy was used to visualize membrane fluctuations of GUVs after addition of complement proteins (top row) or when buffer replaced C5b6 (bottom row). Formation of functional pores allowed passage of sucrose across the bilayer, resulting in loss of contrast (left and right panels). White scale bar, 10 μm. Colored contours depict the mean root-mean-square deviation (RMSD) value of membrane fluctuations for a representative GUV after the addition of complement proteins. Black scale bar for RMSD, 0.5 μm. **b** Individual dots show the RMSD of membrane fluctuations for each GUV normalized the fluctuations after C7 addition. Bar graph represents the mean and is colored according to contour plots in **a**. Error bars represent standard deviations. *P* values comparing the membrane fluctuations of individual GUVs through all steps of MAC formation were calculated using a Wilcoxon match-pairs test (****$P < 0.0001$, *$P < 0.1$). **c** Sequential changes in bending modulus (Δκ) after the addition of C5b6, C7, and C8. Δκ for individual GUVs tracked during MAC formation are indicated (filled circles). Mean Δκ for each step (black bars) and the mean standardized to control vesicles that lack C5b6 (empty circles) are indicated. Error bars represent standard deviations (C5b6, $n = 12$ GUVs; C5b7, $n = 15$; C5b8, $n = 10$). *P* values were calculated using a Mann–Whitney test (***$P < 0.001$) comparing Δκ after each step of MAC assembly with controls. **d** Schematic illustrating how MAC ruptures lipid bilayers. C5b6 (C5b, gray; C6, blue) interacts with the membrane outer leaflet and reduces the activation energy required for bending the bilayer. C7 (orange) binds C5b6 to form the C6b7 complex, which anchors to the membrane and maintains the lowered activation energy. C8 (C8α, light purple; C8β, magenta; C8γ, cyan) joins the assembly to form C5b8. Differences in length and charge properties of the membrane-interacting β-hairpins bend and pierce the bilayer to form a nonlamellar edge of an arc pore (red arrow). Sequential recruitment and insertion of C9 (green) forms MAC and the membrane stiffens

mediated lysis. It will be interesting to understand the interplay between MAC components and its inhibitor within the context of the membrane. In addition to complement proteins, CD59, also acts as a co-receptor for the cholesterol-dependent β-pore forming toxin Intermedilysin[44]. CD59 binds a β-hairpin of the toxin's auxiliary domain that contains the partially inserted cholesterol recognition loops[45]. Intriguingly, we have found that residues of MAC components that interact with CD59[46,47] lie within β-hairpins of C8α and C9. As changes in membrane fluctuations impact diffusion of GPI-anchored proteins[48], we speculate that CD59 may respond to changes in physical properties of the membrane and capture the newly formed β-hairpins of complement proteins during their helix-to-hairpin transition.

Our structural and biophysical data inform a general mechanism for β-pore forming proteins (Fig. 4d). Large-scale rotations of regulatory domains accompany the aqueous-to-transmembrane transition of soluble monomers. We propose that partially inserted proteins initiate interaction with the outer leaflet and prime the membrane for attack by lowering the energy for bending the lipid bilayer, a mechanism exploited by malarial parasites during cell invasion[49] and by the insertion and folding of β-barrel proteins[50]. Subsequent protein conformational re-arrangements distort the lamellar structure of the bilayer, causing it to stiffen and ultimately rupture. Changes in mechanical properties of the lipid environment could explain the nuanced cellular response to membrane damage by both MAC and bacterial pore-forming toxins.

In conclusion, we show that the MAC is a flexible immune pore that interacts with the lipid bilayer in two distinct ways. These interactions govern how MAC initiates membrane-binding and ultimate rupture. Our observations provide a general mechanism for pore formation and explain how MAC function can be tuned to different targets and cellular contexts.

## Methods

**Purification of MAC pores**. A lipid mix consisting of 1,2-dioleoyl-sn-glycero-3-phosphocholine (DOPC) and 1,2-dioleoyl-sn-glycero-3-phosphoethanolamine (DOPE) (6:4 w/w, Anatrace) in chloroform was dried under nitrogen gas and rehydrated in buffer containing 20 mM HEPES-NaOH, 150 mM NaCl at pH 7.4. Rehydrated liposomes were extruded through a 100 nm polycarbonate membrane (Whatman) to produce a monodisperse solution of unilamellar liposomes, whose local membrane curvature is negligible on the length scale of the complete pore. Complement proteins C5b6, C7, C8, and C9 (Complement Technology) were sequentially incubated with liposomes at a molar ratio of 1:1:1:18. A 5 min incubation step was allowed between each component addition followed by incubation for 1 h at 37 °C to optimize for assembly completion before transferring to 4 °C overnight. Assembled MAC complexes were solubilized in 1.5 % Cymal-5 (Anatrace) in the presence of DOPC (1 mg/ml) and glycerol (10 %) for 1 hour at room temperature. Solubilized complexes were purified using density centrifugation in a sucrose solution (5–20%) containing 0.004 % Cymal-7 NG (Anatrace). Samples were spun for 4 hours at 45,000 rpm using an SW60Ti rotor. Fractions were screened using negative stain EM and those containing complete MAC pores were pooled, concentrated and sucrose removed using a ZebaSpin desalting column (Thermofisher Scientific).

**De-glycosylation of C8 and C9**. C8 and C9 were incubated with PNGase F (N-Glycosidase F, Roche) at the ratio 1 μg:0.67 U and diluted in protein buffer (120 mM NaCl, 20 mM Hepes pH 7.4) to a final concentration of 0.3 mg/ml. Under control conditions C8 and C9 were mixed in identical ratios with buffer (50 mM sodium phosphate pH 7.3, 12.5 mM EDTA, 50% glycerol) and diluted to 0.3 mg/ml. Reaction mixes were incubated for 48 h at 37 °C.

**Liposome lysis assay**. A DOPC:DOPE lipid mix (6:4 mol:mol) was resuspended in calcein solution (50 mM calcein, 150 mM NaCl, 20 mM Hepes pH 7.4), freeze-thawed 6 times (−196 °C to 65 °C) and extruded through a 100 nm polycarbonate membrane to form calcein encapsulated liposomes. Liposomes were purified on a gravity flow Sephadex-G50 (GE Healthcare) column to remove non-encapsulated calcein. Calcein is self-quenched at 50 mM and is un-quenched through its dilution in the outer solution following MAC lysis of liposomes. Liposomes lysis was performed by sequential addition of C5b6 (5 min, 37 °C), C7 (5 min, 37 °C), C8 and C9 at a mass ratio of 1:1:1:1. Fluorescence was recorded immediately following C9 addition and every minute for 60 min on a SpectraMax M2 fluorometer (Molecular Devices) with monochromator excitation at 490 nm and emission recorded at 520 nm. For each independent recording, background fluorescence (calcein encapsulated liposomes alone) was measured and subtracted from the data before normalizing to the maximal fluorescence value. Maximal lysis fluorescence was recorded after bursting the liposomes through a freeze-thaw cycle. MAC lysis of liposomes was measured in three independent replicates.

**Negative stain EM of MAC pores on lipid monolayers**. To form lipid monolayers, 4 mm diameter wells in a Teflon plate were filled with 9 μl of buffer (120 mM NaCl, 20 mM Hepes pH 7.4) and overlaid with 2 μl of DOPC:DOPE (60:40 mol:mol) at 1 mg/ml in chloroform. Chloroform was allowed to evaporate for 1 min and a CF400-CU grid (Agar Scientific) was deposited at the surface of the well with the carbon side facing the solution. To form MAC pores, all steps were performed at 37 °C and the Teflon plate was kept on buffer soaked tissue in a closed container to maintain constant humidity. With minimal perturbation to the monolayer C5b6, C7, C8 (60 nM final concentration), and C9 (1.2 μM) were sequentially added to the solution at molar ratios of 1:1:1:20. A 5 min incubation step was allowed between each component addition followed by 15 min incubation after C9 addition. Grids with adherent monolayers were then gently peeled off the solution and directly stained in uranyl acetate 2% (w:v). Images were acquired on a Tecnai F20 electron microscope (Thermo Fisher Scientific) with a Falcon II camera at ×50,000 magnification (2.05 Å/pixel), 0.75–1.5 μm underfocus. Grids were imaged across 4 quadrants of the grid to control for local variations in monolayer composition. For PNGase F digested samples, images were acquired on a Tecnai T12 electron microscope (Thermo Fisher Scientific) with a F216 camera (TVIPS) at ×42,000 magnification (3.71 Å/pixel).

**CryoEM**. In order to obtain a sufficient proportion of intact pores on cryoEM grids, freshly purified solublized MAC (2.5 μl) was applied to glow-discharged holey carbon grids (Quantifoil R 1.2/1.3). Samples were flash frozen in liquid ethane cooled in liquid nitrogen using a Vitrobot Mark III (Thermo Fisher Scientific) and stored under liquid nitrogen until use. Screening of cryoEM conditions was performed on a 120 kV Tecnai T12 (Thermo Fisher Scientific). Eight datasets were collected on 300 kV Titan Krios microscopes (Thermo Fisher Scientific) equipped with Falcon II or Falcon III direct electron detectors (Thermo Fisher Scientific), at a defocus range of 1.75 to 4 μm underfocus. Exposures were recorded as movies comprising of 32–39 frames. Due to the flexibility and low concentration of detergent solubilized pores, large datasets were collected to obtain sufficient populations of homogenous particles. Particle distribution was highly dependent on ice-thickness, therefore only holes that allowed us to obtain both sufficient contrast and monodisperse particles were selected. Some datasets included the use of carbon-coated holey carbon grids to improve the distribution of particles. A summary of imaging conditions is provided in Supplementary Table 1.

**Image processing**. Electron micrograph movie frames were aligned by Motion-Cor2[51], discarding the first and last frames. CTF parameters were estimated using CTFFIND4[52]. Any movies containing low figure of merit scores, substantial drift, low contrast, thick ice, or crystalline ice were discarded from further analysis. Particles were manually selected and extracted from high-quality aligned movies using RELION[53]. Particles were subjected to iterative rounds of 2D classification to improve the homogeneity of the dataset. The published MAC reconstruction was strongly low-pass filtered (60 Å) to prevent model bias and used as a starting model for a gold-standard 3D autorefinement of images. 231,767 selected particles contributed to a consensus MAC structure whose resolution was determined at 4.9 Å. These orientations served as the starting point for tracking beam-induced movement of individual particles, which was corrected using particle polishing within RELION.

3D classification of images revealed conformational heterogeneity of the MAC (Supplementary Fig. 1). Particles containing an intact closed β-barrel (23%) were grouped and subjected to an additional round of 3D autorefinement (5.6 Å resolution). Those that were stoichiometrically identical to the closed conformation but had an open β-barrel (35%) were grouped separately and independently refined (5.6 Å resolution). It was not possible to improve the maps as a whole for the two conformations because of a continuous relative rotation between the asymmetric region (C5b6, C7, C8, and neighboring C9) and the C9 oligomer. We therefore solved the structures of the asymmetric region (4.7 Å and 5.9 Å, open and closed respectively), hinge of the open conformation (4.9 Å), and C9 oligomer of the open conformation (4.4 Å) separately. The global open and closed maps were used as a reference to assemble the parts together. To improve the alignment of the asymmetric and hinge regions, density corresponding to the C9 oligomer was subtracted from the original images. Orientation parameters for the asymmetric and hinge regions were refined from density-subtracted images by applying a mask based on its position in maps generated from the original images. Masks were optimized to include regions of the structure that moved together and were included in subsequent rounds of 3D autorefinement. A similar procedure was adopted to focus the refinement on the C9 oligomer. A final resolution of 4.4 Å was achieved for the C9 oligomer within the open conformation by averaging density for 8 neighboring copies in Chimera[54]. Resolution of all maps was determined using the masking-effect corrected Fourier Shell Correlation (FSC) as implemented in RELION post-processing. Local resolution estimates were calculated within a soft spherical mask that is translated across the map, using phase-randomization to assess the convolution effects of the mask and locally low-pass filtered, as implemented within RELION.

**Model building and refinement**. Crystal structures for soluble C5b6[11], and C8[21], together with homology models for C7 and C9 were used as initial models for refinement into local resolution filtered maps. Due to differences in map quality between the two conformations, the asymmetric and hinge regions of the open state were used for refinement of C5b6, C7, and C8. C9 was refined into the locally averaged C9 oligomer of the open conformation, and placed as a single rigid body in the remaining 17 copies within the map. TMH regions of C6 and C8 were removed from the soluble structures prior to fitting, as these are known to undergo a structural transition to form β-hairpins. Domains of C5b (C345C and MG8), C6 (FIM1 and FIM2), and C7 (FIM1 and FIM2) for which there was weak or no

density were also removed from the model. Truncated crystal structures for C5b6 and C8αβγ were initially placed manually in to the map using Chimera followed by rigid body fitting using the Fit in Map tool. These atomic models were split into domains, which were real-space refined as rigid bodies in Coot[55]. Domains that comprised the primary interaction interface between C5b and C6 were grouped as a single group (C6: CCP1, CCP2, TSP3, and C5b: TED) in the first instance. C5b MG domains were also grouped in the early stages of refinement. As there are no crystal structures available for C7 or C9, homology models were generated using MODELLER[56]. C8β was chosen as template for C9 (26.76% sequence identity), while the C7 model was based on the crystal structure of soluble C6 (31.87% sequence identity) together with reference structures for the C7 Factor I-like domains (PDB:2WCY)[57]. CH3 helices of MAC proteins were remodeled with MODELLER and the 15 C-terminal residues of C9 were manually built in Coot. For C9, the LDL, EGF, and TSP domains were flexibly fitted in a stepwise, iterative process whereby large-scale movements were refined first using iMODFIT[58] followed by further local real-space refinement using Flex-EM[59]. Here, segments of the structure were restrained based either on user-defined rigid bodies or those defined by RIB-FIND[60]. The fitting progress was analyzed by local scoring using the Segment Based Manders' Overlap Coefficient (SMOC)[61], as implemented in the TEMPy software[62]. Refinement was carried out iteratively until the CCC between the map and model stabilized. Once each component was fitted, interfaces were assessed for clashes with Chimera followed by further refinement of the sub-complexes, as necessary. Loops connecting domains were refined in Coot or removed if density was not apparent. Models were further optimized using PHENIX real-space refinement[63] with secondary structure element and disulfide bond restraints. Where crystal structures were available, reference-based restraints were also imposed. The MAC pore β-hairpins were extended with idealized β-strands, in which main-chain geometry was generated using an in-house program to set amino acid (phi, psi) angles to (−140, 135). A full atomic model was then generated by adding side-chains to backbone atoms using the SCWRL4 program[64]. Modeled strands were least squares-fitted onto existing β-strands of core MACPF domains and manually adjusted to minimize clashes using Coot. Overlapping residues were removed. Models were merged and linking residues were real-space refined in Coot. Known glycosylation sites were added to the extended model in Coot, with placement guided both by residue position and visible density. The final models were subjected to a final refinement using global minimization in PHENIX with secondary structure and di-sulfide bond restraints. The quality of the final fits between maps and models per residue was assessed using TEMPy SMOC score (Supplementary Fig. 3). Statistics of overall model quality and geometry outliers for final models were reported using MolProbity[65] (Supplementary Table 1).

**Map visualization and analysis**. Density maps and models were visualized in Chimera. Local resolution of the maps and angular distribution of the particles were assessed in RELION and visualized in Chimera. Coulombic potentials of interaction interfaces were calculated and visualized in Chimera. Maps sharpened with a global B-factor and low pass filtered according to local resolution estimates were used for fitting and refinement. Interaction interfaces and structural re-arrangements of complement proteins were analyzed in Coot. Structural movie and figures were generated in Chimera.

**Flicker spectroscopy**. DOPC:DOPE (60:40 mol:mol) was dissolved at 1 mg/ml in chloroform and coated onto the conductive side of an indium tin oxide coated glass slide (Sigma-Aldrich). Following chloroform evaporation for 30 min, two glass slides (one coated with lipid, one without) were placed either side of a custom-made Polydimethylsiloxane (PDMS) o-ring (Sigma-Aldrich) with the conductive sides of the slides facing inwards to form a chamber. The chamber was filled with a solution of 290 mM sucrose, 1 mM Hepes pH 7.4 and the conductive slides were connected to a TG315 signal generator (Aim-TTi Instruments). To electro-form GUVs, an alternating potential of 1 V at 10 Hz was applied through the slides for 2 h, followed by 1 V at 2 Hz for 1 h to detach the GUVs from the glass. The sucrose containing GUVs were then diluted in a hyperosmotic solution of 1% bovine serum albumin (BSA), 360 mM glucose, 50 mM NaCl, and 10 mM Hepes pH 7.4 and imaged on the day they were produced.

GUVs were imaged in CoverWell perfusion chambers (Grace Bio-Labs) attached to BSA-coated glass slides, allowing the sequential injection of MAC components while continuously tracking individual GUVs. Fluctuation videos were recorded on an Eclipse TE2000-E microscope (Nikon Instruments) at ×30 or ×60 magnification using a Zyla sCMOS camera (Andor) at a frame rate of ~30 frames per second and an exposure of 0.5 ms. MAC formation was achieved through sequential addition of C5b6, C7, C8, and C9 in 1:1:1:21 molar ratios allowing for a 5 min incubation between each addition. Thirtysecond fluctuation videos were recorded prior to C5b6 addition and following the addition and incubation of each component. Loss of GUV contrast was used as an indicator for full MAC deposition and lysis of the GUV, and was observed 8–12 min post C9 addition. GUVs were chosen at random from those that were visibly fluctuating and all GUVs that lysed post-C9 addition were used for analysis (>95% of GUVs). In control experiments, C5b6 was replaced by protein buffer (120 mM NaCl, 10 mM Hepes pH 7.4); the rest of the protocol was identical.

Analysis of bending modulus and tension variations was performed using a custom-built LabView program (National Instruments)[66]. Briefly, GUV contour

coordinates were extracted with subpixel resolution from each video frame and Fourier transformed to extract fluctuation modes. The fluctuation mode amplitudes were averaged across all frames in a particular video to give mean square amplitudes at the GUV equator ($h^2(q_x y = 0)$). These were plotted as a function of the mode wavenumber ($q_x$) and the following model was fitted to the data to extract bending modulus ($\kappa$) and tension ($\sigma$) values as described in equation (1):

$$h(q_x, y = 0)^2 = \frac{1}{L}\frac{k_B T}{2\sigma}\left(\frac{1}{q_x} - \frac{1}{\sqrt{\frac{\sigma}{k_c} + q_x}}\right) \tag{1}$$

where $k_B$ is the Boltzmann constant, $T$ temperature, and $L$ the mean GUV contour circumference. Fluctuation data were fitted from mode 4 to mode 20 (Supplementary Fig. 7b). Failure of fits were mostly a result of contrast loss due to GUV lysis after C9 addition, bending modulus variations were hence never extracted for this step. In some cases, fitting was not possible at earlier steps of MAC formation as a result of poor contour extraction, due to image interference caused by neighboring GUVs or significant deformation of the GUV leading to unreliable contour tracking. For these GUVs, bending and tension were not extracted at the step of fit failure but were analyzed at previous steps and integrated into the data set. In addition, changes in the extent of fluctuation in each GUV were quantified following the addition of each MAC component by calculating the contour RMSD (averaged over all frames for each video).

For all flicker spectroscopy experiments, 18 individual GUVs were tracked through MAC formation over three independent experiments. By measuring changes in bending modulus across the same vesicle throughout the assembly process, these measurements are independent of intrinsic small variations in GUV mechanical properties within a population of vesicles. Control experiments were performed on the same day and same batch of GUVs. As some data sets did not follow a normal distribution, as defined by the D'Agostino & Pearson and Shapiro–Wilk tests, all variation significances were assessed with double-sided non-parametric tests (Mann–Whitney and Wilcoxon match-pair tests). All statistics were computed using Prism (GraphPad Software), figure plots were generated using DataGraph (Visual Data Tools).

**Code availability**. Custom software for membrane fluctuation analysis will be provided upon request.

## Data availability
CryoEM data and corresponding atomic models have been deposited in public repositories. Seven maps have been deposited in the Electron Microscopy Data Bank with accession codes: EMD-0106, EMD-0107, EMD-0109, EMD-0110, EMD-0111, EMD-0112, EMD-0113. Atomic coordinates have been deposited in the Protein Data Bank with accession codes PDB: 6H03, 6H04. Other data are available from the corresponding author upon reasonable request. A reporting summary for this Article is available as a Supplementary Information file.

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

## Acknowledgements

We thank A. Carter for discussions; D. Clare, S. Welch, A. Seibert, C. Hecksel, and F. de Haas for data acquisition assistance; S. Islam for computational support, and J. Lo for help in selection of particles. We thank the LMB-MRC for access to electron microscopes and Diamond Light Source for access to eBIC (proposals EM19429, EM13304, EM16919, EM13893, EM12204, and EM12388) funded by the Wellcome Trust, MRC and BBSRC. This work is supported by a CRUK Career Establishment Award (C26409/A16099) to D.B.; C.M.B. is funded by a BBSRC Doctoral Training Program grant, Ref: BB/J014575/1;

N.J.B. is supported by a EPSRC Programme Grant (EP/ J017566/1); A.P.J. and M.T. are supported by MRC (MR/M019292/1).

## Author contributions

M.S. and A.M. prepared samples. M.S., A.M., C.M.B., and D.B. performed the electron microscopy. M.S., A.M., D.B., C.M.B., and S.G. contributed to the image processing. M.S., C.M.B., A.P.J., D.B., and M.T. performed molecular modeling. A.M. and N.J.B. performed flicker spectroscopy. B.P.M. contributed to discussions and analysis of MAC structure. All authors contributed to the experimental design, data analysis, and preparation of the manuscript. M.S. and C.M.B. contributed equally.

## Additional information

**Competing interests:** The authors declare no competing interests.

