## [Peer Review File · Nature Communications]

Reviewers' comments:

Reviewer #1 (Remarks to the Author):

This paper by Doryen Bubeck and colleagues describes a subtle, thoughtful, detailed and careful analysis of two different conformations of the membrane attack complex (MAC) of the complement system. The paper provides a description of domain rearrangements associated with the stepwise assembly of this complicated system for the targeted disruption and perforation of membranes and describes data yielding very interesting insights into the biophysics of membrane interactions by, e.g. peripheral as well as pore-forming proteins.

Among the many strengths of this paper are the ways in which the authors use the fitting of structures into their 3DEM density to infer the mechanical processes in MAC pore formation, the way in which they make well-judged and well-described use of sub-averaging to bring different regions of the assembly into sharper focus, and the insights they provide into the ways in which the individual MAC components contribute to its overall assembly and function - for example, via TMH regions of differing lengths, by perturbing the lipid packing in the membrane under attack, by indicating how their findings suggest a mechanism for facilitating the formation of the lipid structures associated with toroidal pores, in the way that C8alpha and C9 interact with CD59, etc.

This paper is the highest resolution study of the MAC in a functional state we have. It describes a significantly more important set of insights than could be provided by a poly(C9) structure, for example, which is not physiologically relevant. The clear resolution of two different MAC states related by a breathing motion is a significant step forward in our understanding of MAC assembly and activity.

I only have minor requests for changes.

At the end of the Introduction and when it is later discussed I think it would be good to have a but more description about flicker spectroscopy, for the less informed reader.

At lines 190-192 the authors suggest ways in which a structural role for glycans in generating a more ordered assembly could be biologically necessary for the passage of proteins that degrade peptidoglycan layers. I wondered if a simpler or more basic explanation might just be that the N-linked glycans in question confer greater robustness on the pore assembly which might be of benefit in the physiological context for a whole host of reasons - resisting sheer forces perhaps or to prevent proteolysis. In any case they show that the polypeptide core is sufficient for pore formation in vitro and clearly the selection of glycosylation in vivo will be related to benefit.

At lines 215-217 the authors suggest that equilibration of extracellular milieu across the pore could give rise to membrane rigidification: but couldn't this also be simply due to the membrane insertion of the MAC? This too would rigidify the membrane, I think?

Overall, the authors are to be congratulated on a fine piece of work.

Reviewer #2 (Remarks to the Author):

The authors reported a cryo-EM structure of the MAC pore with a resolution of 8.5 Angstrom in 2016. Now, with a larger dataset, the resolution was improved to about 5 Angstrom and two conformations were observed. Intuitively, one may think the closed conformation is more stable than the open conformation in the presence of a lipid membrane instead of the open conformation (what will be used to fill the opening on the wall of the pore?). Is the open conformation just the artifact due to the lack of lipid membrane? Is there any biochemistry or other evidence to show that the open conformation exists in vivo and deserve all focus of this manuscript?

Another major concern is the reliability of the atomic model. In the resolution about 5 Angstrom, side chains cannot be identified, and only the helices and sheets can be identified. So it is possible that some amino acids are shifted or misaligned. Only the CC score between the map and the model is not enough. More details need to be given.

The importance of glycosylation is over-emphasized. The deglycosylation only changes the pore curvature, but not the activity. Furthermore, will the claimed density of T427 and N394 be still visible when the contour level was increased to see the separation of helices in supplementary Fig 5A?

As there are no lipid bilayers in the structure, the position of the aromatic residues may not at the headgroup of lipid molecules. Furthermore, the terminal C9 and the C9 next to C8 are shifted by about one membrane thickness, which brings some aromatic residues to be at the bilayer center or out of the bilayer. What makes the aromatic residues travel across the claimed lipid membrane? Is this the indication of artefact due the lack of lipid membrane?

Focused alignment was applied to the open conformation. Why was not applied to the closed conformation?

Masked refinement from signal-subtracted images followed by sub-volume averaging was used to resolve the averaged C9 monomer at 4.4 Angstrom (Fig. 1a, Supplementary Figs. 1, and 2b). However, the FSC curve in supplementary Fig 2b shows a plateau around 5 Angstrom, which is usually due to the mask used in the FSC calculation. Is there over evidence that the resolution is improved?

In the paragraph before section MAC glycan scaffold, the electrostatic interaction was discussed. However, it is not clear whether the positive patch is in contact with the negative patch in Fig 2C.

In page 6 and supplementary Fig 3a, the C5b6 structure was claimed to be membrane associated. However, in the purification MAC pores, 1.5% of Cymal-5 was used in the presence of 1mg/ml DOPC, which means only tightly-bound lipid molecules would survive the purification. So it is not membrane associated.

In Fig 3C, there are three conformations of the native MAC with a fraction of 41%, 38% and 14%. In supplementary Fig 1, the closed beta-barrel is 23% while the open beta-barrel is 35%. Is there any relationship between the conformations observed in negative stain EM and those observed in cryoEM? Also, why does the 23% and 35% of particles give the same resolution (5.6 angstrom)? What other conformations were observed in cryoEM for the 42% of the particles?

Why the classes shown in Fig 3C and supplementary Fig 2A are different?

In the purification of MAC pores, it says that complement proteins C5b6, C7, C8, and C9 (Complement Technology) were sequentially incubated with liposomes at a molar ratio of 1:1:1:18 for 1 hour at 37 °C. Does it mean waiting for one hour after adding each component? Or does it mean waiting for one hour after all the components were added? In addition, how the density gradient was designed is not clear.

In Map visualization and analysis section, it says that maps sharpened with a global B-factor and low pass filtered according to local resolution estimated were used for fitting and refinement. Was the local resolution for each voxel or each subdomain? How was that performed? Or how were the filtered maps assembled?

Reviewer #3 (Remarks to the Author):

The manuscript entitled „CryoEM reveals how the complement membrane attack complex rupture lipid bilayers” by A. Menny et al. propose, based on experimental evidences, supplemented with advanced image analysis methods and model building, the mechanism of membrane associated complement proteins assembly leading to membrane destabilisation. The excellent methodology used for data acquisition and generation of images make possible to resolve the pore structure and propose the mechanistic description of the pore formation process. In addition, the manuscript contributes to the understanding of beta barrel formation in the lipid matrix, which is still not well understood.

However, in order to make arguments stronger, the following issues should be address.

- 1) The selection of lipid matrix may imply that the hydrophobic effect is a major driving force for the pore formation processes. The selected experimental model of the lipid bilayer (DOPC:DOPE; 6:4 w/w) does not account for the electrostatic effect. Despite the fact, that it is well established that the bacteria membranes contain significant proportion of negatively charged lipids (i.e. PG). Authors should address the potential role of electrostatics as a comment for the selection of the experimental model.
- 2) The complementarity between experimental models used: LUV, GUV and supported membranes need to discuss in length stressing potential sources of discrepancies.
- 3) The sample preparation methodologies for each experimental technique are qualitatively different. Whereas the reconstitution of the protein pores in LUV model is reliable, since a well-defined subpopulation of vesicles are selected for subsequent analysis. In a case of the two other models it is not so obvious. Protein aggregates formed on the supported membranes or GUV vesicles are likely heterogeneous. The image selection criteria are not so obvious and a resulting data are prone to biases. In the first case the criteria for the exclusion of image should be clearly and if possible quantitatively stated. The GUV model is also challenging, since it is very difficult, if not impossible, to obtain the uniform vesicle population, especially from mixture of lipids. The vesicle selection criteria should be clearly stated, such as the vesicle size distribution and/or vesicle composition. When calculating the lipid bilayer bending rigidity based on the small population of vesicles (15 or so) it is necessary to demonstrate that the normal distribution requirement is satisfy, see for example [J. Doskocz et al. J. Membr. Biol. 2018 Aug; 251(4):601-608], so the statistical analysis used is justifiable.
- 4) The correlation between the local membrane event, such as the protein pore formation, and global properties, such as bending rigidity coefficient change is interesting but very speculative, since the correlation is likely concentration-dependant (depending on the number of pores (proteins) present on/in a single membrane). It would improve the manuscript if the issue would be discussed in depth.

There are few small editorial mistakes as in the Result section line forth. There is 100 nM but should be 100 nm.

Summarizing, the manuscript is a valuable contribution to the understanding of the formation and functioning of multicomponent complex. However, the relative balance between protein and lipid components of the complex has to be at address to grater extend.

I recommend the manuscript for publication after issues listed above are addressed.

Reviewer #1

Comment 1

At the end of the Introduction and when it is later discussed I think it would be good to have a bit more description about flicker spectroscopy, for the less informed reader.

Response 1

This is now done.

“To investigate how these two types of interactions impact mechanical properties of the target cell membrane, we performed fluctuation analysis (flicker spectroscopy) of giant unilamellar vesicles (GUVs). In flicker spectroscopy, the magnitude of thermally induced fluctuations in a GUV membrane are quantified by tracking the membrane contour in a series of video-microscopy images. This data is fitted to a two-parameter model to extract the membrane bending rigidity and tension³².”

Comment 2

At lines 190-192 the authors suggest ways in which a structural role for glycans in generating a more ordered assembly could be biologically necessary for the passage of proteins that degrade peptidoglycan layers. I wondered if a simpler or more basic explanation might just be that the N-linked glycans in question confer greater robustness on the pore assembly which might be of benefit in the physiological context for a whole host of reasons - resisting sheer forces perhaps or to prevent proteolysis. In any case they show that the polypeptide core is sufficient for pore formation in vitro and clearly the selection of glycosylation in vivo will be related to benefit.

Response 2

We have now simplified our discussion of glycans in the manuscript and moved it to support the section regarding MAC flexibility. Figure 3 has now been moved up to be Figure 2. The supporting figure in the supplement has been moved up to Supplementary Fig. 4.

“The MAC is a highly-glycosylated assembly with all complement components post-translationally modified^{11,25-27}. We observe density for many of the reported glycans on C7²⁵, C8²⁷ and C9²⁶, which line the β -barrel's concave face (Figs. 1b-c, 2a and Supplementary Fig. 4). Glycan removal led to irregular pores with significantly distorted curvature (Supplementary Fig. 4). These data suggest glycans could play a role in maintaining the structural integrity of a flexible giant β -barrel. Although deglycosylation did not impact the ability to rupture simple model membranes (Supplementary Fig. 4), glycans may confer greater robustness on pore assembly.”

Comment 3

At lines 215-217 the authors suggest that equilibration of extracellular milieu across the pore could give rise to membrane rigidification: but couldn't this also be simply due to the membrane insertion of the MAC? This too would rigidify the membrane, I think?

Response 3

This has now been included.

"These data suggest that large MAC lesions in cells may lead to rigidification of the membrane by the insertion of β -hairpins across the bilayer and by equilibration of the extracellular milieu across the pore."

Reviewer #2

Comment 1

Intuitively, one may think the closed conformation is more stable than the open conformation in the presence of a lipid membrane instead of the open conformation (what will be used to fill the opening on the wall of the pore?). Is the open conformation just the artifact due to the lack of lipid membrane? Is there any biochemistry or other evidence to show that the open conformation exists in vivo and deserve all focus of this manuscript?

Response 1

Arc pores, in which lipid fills the opening on the wall of the pore, is a well-documented phenomenon for β -pore forming proteins. Arc pores have been observed for both mammalian and bacterial pores and have been visualized within the context of a lipid bilayer using electron cryo tomography (Sonnen, et al. 2014) and atomic force microscopy imaging (Leung et al., 2017).

Sonnen A., Plitzko JM, Gilbert RJC. Incomplete pneumolysin oligomers form membrane pores. *Open Biology*. 2014 Apr 23;4:140044. doi: 10.1098/rsob.140044.

Leung C, Hodel AW, Brennan AJ, Lukoyanova N, Tran S, House CM, Kondos SC, Whisstock JC, Dunstone MA, Trapani JA, Voskoboinik I, Saibil HR, Hoogenboom BW. Real-time visualization of perforin nanopore assembly. *Nature Nanotechnology* 2017 Feb 06;12, 467-473. doi:10.1038/nnano.2016.303.

We have now included the following statement to reference these two publications.

"Lipid molecules likely fill the opening on the wall of the pore, reminiscent of arc pores observed for both mammalian and bacterial β -pore forming proteins^{22,23}."

Comment 2

Another major concern is the reliability of the atomic model. In the resolution about 5 Angstrom, side chains cannot be identified, and only the helices and sheets can be identified. So it is possible that some amino acids are shifted or misaligned. Only the CC score between the map and the model is not enough. More details need to be given.

Response 2

Although density is lacking for many side-chains within C5b and the lower half of the central β -barrel, we have a large number of experimental restraints that have been included in the model building and refinement that justify their register within the atomic model. Crystal structures are available for C5b, C6, and C8 components of the MAC. Domains of these structures were first refined as rigid bodies, while enforcing disulfide bond restraints. Subsequently, models were refined imposing structural

restraints (side-chain and secondary structure) to the higher resolution crystallographic structures. Strands that comprise the central β -barrel were initiated where side-chain density was visible and then extended with geometric backbone constraints for an idealized β -strand and were not refined further. The trajectory for each strand is linear and the register was confirmed by the correlation of glycan density with the position of modified residues in the sequence.

In addition to the details already provided in the methods section. We now include a short summary of the model building in the main text.

“Although density is lacking for many side-chains within C5b and the lower half of the central β -barrel, we have imposed experimental restraints that justify their register in the atomic model. Crystal structures for soluble components (C5b6¹¹ and C8²¹, α heterotrimeric complex consisting of α , β , and γ polypeptide chains) together with homology models for C7 and C9 were fitted into the density. Domains of these structures were first refined as rigid bodies, with disulfide bond restraints. Models were further refined restraining secondary structure and side-chain geometry to higher resolution crystallographic structures. β -Strands that comprise the central barrel were initiated where side-chain density was visible and extended imposing idealized backbone geometry constraints. The trajectory for each strand is linear and the register was confirmed by correlating glycan density with the position of the modified residue in the sequence (Supplementary Fig. 4).”

We agree with Reviewer 2 that a global cross correlation score of model to a map whose local resolution widely varies is not informative. The SMOG scoring of the model on local resolution filtered maps offers a per residue assessment within their local environment. Initially we scored the complete model for the pore for the global reconstruction for each conformation. The resolution of these maps was lower than that of the subvolume reconstructions where more side-chain density was visible. We now include SMOG scores of models within the subvolume maps and images of the map and model overlay showing clear density for side-chains where the β -hairpins initiate and terminate. Details can be found in a new Supplementary Figure 3.

Comment 3

The importance of glycosylation is over-emphasized. The deglycosylation only changes the pore curvature, but not the activity. Furthermore, will the claimed density of T427 and N394 be still visible when the contour level was increased to see the separation of helices in supplementary Fig 5A?

Response 3

In light of comments from Reviewer 1 and 3, we have simplified our discussion and interpretation of MAC glycosylation. Please see response to Reviewer 1, comment 2. In doing so, we have removed our discussion of T427 and have removed supplementary Fig 5A.

Comment 4

As there are no lipid bilayers in the structure, the position of the aromatic residues may not be at the headgroup of lipid molecules. Furthermore, the terminal C9 and the C9 next to C8 are shifted by about one membrane thickness, which brings some aromatic residues to be at the bilayer center or out of the bilayer. What makes the aromatic residues travel across the claimed lipid membrane? Is this the indication of artefact due the lack of lipid membrane?

Response 4

Although we do not directly observe the position of the lipid bilayer in our experiment, the positions of aromatic residues have been used to demark the location of the membrane for a related detergent-solubilized β -pore forming protein (Borkori-Brown M., et al., 2016). Furthermore, it is not our intention to suggest that the aromatic residues travel across a static bilayer, rather more likely is that the dynamic nature of the assembly impacts the planarity of the lipid bilayer.

Borkori-Brown M, Martin TG, Naylor CE, Basak AK, Titball RW, and Savva CG. Cryo-EM structure of lysenin pore elucidates membrane insertion by an aerolysin family protein. *Nature Communications*. 2016 Apr 6;7:11293. doi: 10.1038/ncomms11293.

We have added the following lines and reference to the Results section to clarify our interpretation.

“Similar to other β -barrel pore assemblies²⁹, two rings of aromatic residues separated by the width of the bilayer anchor C8 and C9 within the membrane (Fig. 2a).”

“In the open conformation, the first and terminal C9 are latterly shifted by approximately 20 Å, while the ring of the closed conformation remains planar (Fig. 1). Therefore, conformational flexibility of the assembly may impact local curvature of the membrane.”

In the Discussion section “motion” has been changed to “rotation”

“We propose that rotation of pore β -hairpins within the bilayer could provide an additional level of membrane destabilization that contributes to lytic activity of β -pore forming proteins.”

Comment 5

Focused alignment was applied to the open conformation. Why was not applied to the closed conformation?

Response 5

Focused alignment strategy was applied to the asymmetric and C9 oligomeric components of both open and closed conformation. However, the resolution of the closed conformation's C9 oligomer could only be resolved to 5.9 Å. As the highest resolution for C9 was achieved for the open conformation, we used this map to build the C9 model.

We have now clarified this in the text.

“A similar analysis of the C9 oligomer from the closed conformation resulted in a lower resolution map. Therefore, we focused our interpretation of C9 on density derived from the open conformation oligomer.”

Comment 6

Masked refinement from signal-subtracted images followed by sub-volume averaging was used to resolve the averaged C9 monomer at 4.4 Angstrom (Fig.1a, Supplementary Figs. 1, and 2b). However, the FSC curve in supplementary Fig 2b shows a plateau around 5 Angstrom, which is usually due to the mask used in the FSC calculation. Is there over evidence that the resolution is improved?

Response 6

The local resolution of the density corresponding to C9 improved from a range of 9.0-4.5 Å to 6.0-4.0 Å after signal-subtraction, focused refinement and subvolume averaging. There are also visible improvements to the density in this area. We now show this in Supplementary Figure 3c.

Comment 7

In the paragraph before section MAC glycan scaffold, the electrostatic interaction was discussed. However, it is not clear whether the positive patch is in contact with the negative patch in Fig 2C.

Response 7

We have now clarified our discussion of electrostatic surface complementarity in the text. We have changed “charge complementarity”, which refers to specific salt bridges formed at an interface, to “electrostatic surface potentials”, which can have a more diffuse effect on protein-protein interactions. We have also quantified the percentage of polar, charged, and hydrophobic residues that comprise the interaction interfaces shown in Fig. 3c (new numbering) and included them in a new Supplementary Table 3. In addition, we have softened our language in the text.

“To investigate the molecular basis underpinning the directionality of MAC’s sequential assembly, we compared electrostatic surface potentials of complement proteins (Fig. 3c). The interface between C6-C7, C8 α -C9, and C9-C9 is formed mainly by polar and charge interactions (Supplementary Table 3). Incorporation of C8 α exposes a negatively charged patch that may influence recruitment of the positively charged face of C9. C9 oligomerization could be propagated by electrostatic complementarity between leading and lagging faces, consistent with CDC pore formation ²⁹.”

Comment 8

In page 6 and supplementary Fig 3a, the C5b6 structure was claimed to be membrane associated. However, in the purification MAC pores, 1.5% of Cymal-5 was used in the presence of 1mg/ml DOPC, which means only tightly-bound lipid molecules would survive the purification. So it is not membrane associated.

Response 8

We thank the reviewer for the opportunity to clarify this point. It was not our intention to claim purification the C5b6 MAC precursor complex from liposomes. Rather, our analysis of membrane-associated C5b6 stems from our MAC-incorporated C5b6 complex. We have now changed “Membrane-associated” to “MAC-incorporated” in Supplementary Fig 5a (new numbering) and in page 7 of the text.

Comment 9a

In Fig 3C, there are three conformations of the native MAC with a fraction of 41%, 38% and 14%. In supplementary Fig 1, the closed beta-barrel is 23% while the open beta-barrel is 35%. Is there any relationship between the conformations observed in negative stain EM and those observed in cryoEM?

Response 9a

The relationship between detergent-solubilized pores visualized in cryoEM and those imaged in their membrane environment using negative stain is shown in Supplementary Figure 2a. In this experiment projections of the two conformations are compared with

reference-free aligned 2D class averages of negatively stained pores in lipid monolayers. The aim of this experiment was to include as many full rings in monolayers as possible to distinguish subtle differences in these conformations over the inherent irregularity of stain granules. Figure 2c is a direct comparison of native MAC and deglycosylated MAC using the equal numbers of particles and groups to enable a more direct assessment of irregularities in curvature.

Comment 9b

Also, why does the 23% and 35% of particles give the same resolution (5.6 angstrom)?

Response 9b

We have thoroughly checked our refinement and postprocessing runs for these two subsets and the resolution value using the FSC cut-off of 0.143 is indeed the same; however the resolution curves themselves do differ. We have re-run refinement with different starting models (all low pass filtered to 60 Å) and with strongly lowpass filtered solvent masks; all converge on the same reconstructions. Postprocessing was performed using a range of soft masks that varied in initial binarisation thresholds, mask extensions, and mask edges. In each case we report the highest resolution estimate after post-processing. Mask-induced artefacts can be diagnosed by plotting the phase-randomized FSC. For each reconstruction, our phase-randomized FSC is below the recommended threshold of 0.1 at the given resolution estimate (see figure below for details). The resolution values for the two maps could be due to different extents of heterogeneity still present in the subpopulations. As such, we have also explored refinement of the open conformation with fewer particles, and as expected the resolution value is lower.

Comment 9c

What other conformations were observed in cryoEM for the 42% of the particles?

Response 9c

The maps resulting 3D classification are now included in Supplementary Fig. 1.

Comment 10

Why the classes shown in Fig 3C and supplementary Fig 2A are different?

Response 10

See response 9. This has been clarified in the legend for Figure 2 (new numbering).

“Classes are derived from similar particle numbers to enable a more direct comparison of curvature distortion (native, 706; de-glycosylated, 711). Native MAC particles are a randomly chosen subset of the data shown in Supplementary Fig. 2a which include

variations across open and closed conformations.”

Comment 11

In the purification of MAC pores, it says that complement proteins C5b6, C7, C8, and C9 (Complement Technology) were sequentially incubated with liposomes at a molar ratio of 1:1:1:18 for 1 hour at 37 ° C. Does it mean waiting for one hour after adding each component? Or does it mean waiting for one hour after all the components were added? In addition, how the density gradient was designed is not clear.

Response 11

This is now clarified in the methods.

“A 5 min incubation step was allowed between each component addition followed by incubation for 1 hour at 37 °C to optimize for assembly completion before transferring to 4 °C overnight.”

“Solubilized complexes were purified using density centrifugation in a sucrose solution (5-20%) containing 0.004 % Cymal-7 NG (Anatrace). Samples were spun for 4 hours at 45,000 rpm using an SW60Ti rotor, as described previously¹⁵.”

Comment 12

In Map visualization and analysis section, it says that maps sharpened with a global B-factor and low pass filtered according to local resolution estimated were used for fitting and refinement. Was the local resolution for each voxel or each subdomain? How was that performed? Or how were the filtered maps assembled?

Response 12

Local resolution was estimated within a small soft, spherical mask that is moved over the entire map, while using phase-randomisation to estimate the convolution effects of that mask, as described in the RELION pipeline. This has now been clarified in the methods section.

“Local resolution estimates were calculated within a soft spherical mask that is translated across the map, using phase-randomization to assess the convolution effects of the mask as implemented within RELION.”

Reviewer #3

Comment 1

The selection of lipid matrix may imply that the hydrophobic effect is a major driving force for the pore formation processes. The selected experimental model of the lipid bilayer (DOPC:DOPE; 6:4 w/w) does not account for the electrostatic effect. Despite the fact, that it is well established that the bacteria membranes contain significant proportion of negatively charged lipids (i.e. PG). Authors should address the potential role of electrostatics as a comment for the selection of the experimental model.

Response 1

There is no chemical specificity of MAC for certain lipids. MAC pores form on a wide variety of membranes spanning negatively charged Gram-negative bacteria as well as nucleated eukaryotic cells. Given the diversity of membrane targets, our experimental lipid composition was optimized for structural homogeneity of pores.

We have included a statement in the Results section to explain the rationale for our experimental model membrane system.

“The human MAC pore was formed on liposomes from individual complement proteins. The lipid composition of vesicles was selected based on the stoichiometric homogeneity of deposited pores. MAC was solubilized with detergent and purified for structural studies, as described previously¹⁵.”

Comment 2

The complementarity between experimental models used: LUV, GUV and supported membranes need to discuss in length stressing potential sources of discrepancies.

Response 2

We used three experimental model systems in this study: 1) LUVs for fluorescence release assays and cryoEM structural analysis, 2) GUVs to parameterize membrane fluctuations, and 3) lipid monolayers for negative stain electron microscopy. Pores formed in LUVs and in lipid monolayers were structurally identical (Supplementary Figure 2a). LUVs were formed by extrusion through a membrane with 100 nm holes. All systems offer an effectively flat membrane environment with respect to the 0.1 nm diameter of the pore itself.

We have now clarified this in the text:

“Rehydrated liposomes were extruded through a 100 nm polycarbonate membrane (Whatman) to produce a monodisperse solution of unilamellar liposomes, whose local membrane curvature is negligible on the length scale of the complete pore.”

Comment 3a

The sample preparation methodologies for each experimental technique are qualitatively different. Whereas the reconstitution of the protein pores in LUV model is reliable, since a well-defined subpopulation of vesicles are selected for subsequent analysis. In a case of the two other models it is not so obvious. Protein aggregates formed on the supported membranes or GUV vesicles are likely heterogeneous. The image selection criteria are not so obvious and a resulting data are prone to biases. In the first case the criteria for the exclusion of image should be clearly and if possible quantitatively stated.

Response 3a

MAC pores formed on LUVs were detergent solubilized and subjected to biochemical purification through density centrifugation to improve sample homogeneity. The pores formed on monolayers were used to show that the two conformations observed after detergent solubilization could be observed in a lipid environment. Therefore, our analysis of pores formed on lipid monolayers does not require a discussion of incomplete assemblies. We agree with the reviewer that more details regarding the image acquisition and quantification of images that went into our 2D EM analyses should be included.

We have included the following statements in the methods section and legend for Supplementary Fig. 2a to detail the image selection criteria for MAC formed on lipid monolayers.

“Grids were imaged across 4 quadrants of the grid to control for local variations in monolayer composition.”

“Classes are derived from 6889 complete pores. Percentage of particles belonging to each class is indicated.”

Comment 3b

The GUV model is also challenging, since it is very difficult, if not impossible, to obtain the uniform vesicle population, especially from mixture of lipids. The vesicle selection criteria should be clearly stated, such as the vesicle size distribution and/or vesicle composition. When calculating the lipid bilayer bending rigidity based on the small population of vesicles (15 or so) it is necessary to demonstrate that the normal distribution requirement is satisfied, see for example [J. Dosekoc et al. J. Membr. Biol. 2018 Aug;251(4):601-608], so the statistical analysis used is justifiable.

Response 3b

Although we recognise that there may be some variation in lipid composition and size of GUVs formed by electroformation, membranes were only composed of DOPC and DOPE, which have similar hydrophobic interactions. Therefore, we expect the compositions of GUVs to be fairly uniform. Furthermore, our analysis minimizes sensitivity to variations in mechanical properties of bare GUVs while maintaining sensitivity to changes induced by pore formation.

In the methods section we now clearly state the selection criteria for vesicles and discuss how our measurements are not sensitive to intrinsic small variations in GUV mechanical properties.

“GUVs were chosen at random from those that were visibly fluctuating and all GUVs that lysed post-C9 addition were used for analysis (> 95% of GUVs).”

“By measuring changes in bending modulus across the same vesicle throughout the assembly process, these measurements are independent of intrinsic small variations in GUV mechanical properties within a population of vesicles.”

As some of our datasets data did not follow a normal distribution, as defined by the D'Agostino & Pearson and Shapiro-Wilk tests, we used non-parametric tests to determine significance of P values. The Wilcoxon match pairs test was used for assessing the significance of membrane fluctuations of individual GUVs. The Mann-Whitney test was used to determine the significance of changes in bending modulus.

We have now stated this in the methods section.

“As some data sets did not follow a normal distribution, as defined by the D'Agostino & Pearson and Shapiro-Wilk tests, all variation significances were assessed with non-parametric tests (Mann-Whitney and Wilcoxon match-pair tests).”

Comment 4

The correlation between the local membrane event, such as the protein pore formation, and global properties, such as bending rigidity coefficient change is interesting but very speculative, since the correlation is likely concentration-dependant (depending on the number of pores (proteins) present on/in a single membrane). It would improve the manuscript if the issue would be discussed in depth.

Response 4

MAC formation is localized to target membranes by the C5 convertase, which proteolytically cleaves C5 to initiate the assembly pathway *in vivo*. Our experimental

system uses a preactivated C5b6 complex to initiate assembly on membranes. As a result, the amount of MAC pores deposited on vesicles is stochastic and not dependent on activation of upstream complement pathways. Our experiments therefore assume that pores are distributed evenly across vesicles.

We now include discussion of how changes in the bending modulus coefficient could be influenced by pore concentration *in vivo*.

“While our experimental system assumes a uniform distribution of pores on GUVs, MAC formation on target cells is heavily influenced by activation of upstream complement pathways and opsonization of bacterial cells. These deposition hotspots may therefore influence changes in bending rigidity coefficients in a concentration-dependent manner.”

Comment 5

There are few small editorial mistakes as in the Result section line forth. There is 100 nM but should be 100 nm.

Response 5

This is now changed.

Reviewer #2 (Remarks to the Author):

As claimed in text, the first and terminal C9 are latterly shifted by approximately 20 Å, while the ring of the closed conformation remains planar (Fig. 1), and the positions of aromatic residues have been used to demark the location of the membrane for a related detergent-solubilized β -pore forming protein. If the aromatic residues marks the position of the membrane, how do the membrane on the two sides of the opening connect? Does it form a ridge like feature?

Response 12: "Local resolution estimates were calculated within a soft spherical mask that is translated across the map, using phase-randomization to assess the convolution effects of the mask as implemented within RELION."

It is clear how the local resolution was determined in Relion. But it is still not clear how the low pass filtering according to local resolution was carried out. Does it mean low pass filter at 4.4 Å? Does it mean the filtering is the same for the entire map?

Reviewer #2

Comment 1

As claimed in text, the first and terminal C9 are latterly shifted by approximately 20 Å, while the ring of the closed conformation remains planar (Fig. 1), and the positions of aromatic residues have been used to demark the location of the membrane for a related detergent-solubilized β -pore forming protein. If the aromatic residues marks the position of the membrane, how do the membrane on the two sides of the opening connect? Does it form a ridge like feature?

Response 1

While the local contour of the membrane is likely impacted by the conformational flexibility of the MAC, we do not feel it is appropriate to further speculate on how lipids are re-ordered at the interface of our detergent-solubilized protein complex.

Comment 2

It is clear how the local resolution was determined in Relion. But it is still not clear how the low pass filtering according to local resolution was carried out. Does it mean low pass filter at 4.4 Å? Does it mean the filtering is the same for the entire map?

Response 2

A new feature of the 'local resolution' job in RELION-2.0 is to output a map that is filtered according to the local resolution estimates (Fernandez-Leiro & Scheres, 2017). Given the flexibility of the MAC, it is not appropriate to filter the maps according to a single overall resolution value; therefore, every map shown in our figures has been filtered according to the local resolution as implemented in the standard RELION-2.0 pipeline. We have now changed the text to explicitly describe how the maps were filtered.

"Local resolution estimates were calculated within a soft spherical mask that is translated across the map, using phase-randomization to assess the convolution effects of the mask and locally low-pass filtered, as implemented within RELION."

Fernandez-Leiro R & Scheres SHW. A pipeline approach to single particle processing in RELION. Acta Crystallogr D Struct Biol. 2017 Jun 1;73(Pt 6):496-502. doi: 10.1107/S2059798316019276.

In the light of these revisions we wish to re-submit our manuscript to Nature Communications. Thank you for your assistance in this.